# Renal Impairment Impact and Survival Analysis in a Romanian Cohort of HIV-1(F1)-Infected Children and Adolescents

**DOI:** 10.3390/life13040888

**Published:** 2023-03-27

**Authors:** Raluca Isac, Rodica Costa, Mirela Frandes, Voichita Elena Lazureanu, Ramona Florina Stroescu, Ruxandra Maria Steflea, Iulia-Cristina Bagiu, Florin George Horhat, Gratiana Nicoleta Chicin, Avram Cecilia Roberta, Pacurari Alina Cornelia, Gabriela Doros, Mihai Gafencu

**Affiliations:** 1Pediatric Department, “Victor Babes” University of Medicine and Pharmacy, Eftimie Murgu Square No. 2, 300041 Timișoara, Romania; 2Emergency Hospital for Children “Louis Turcanu”, Iosif Nemoianu Street, No. 2, 300011 Timișoara, Romania; 3Department of Functional Sciences, Discipline of Biostatistics and Medical Informatics, “Victor Babes” University of Medicine and Pharmacy, Eftimie Murgu Square No. 2, 300041 Timișoara, Romania; 4Department of Infectious Diseases, “Victor Babes” University of Medicine and Pharmacy, Eftimie Murgu Square No. 2, 300041 Timișoara, Romania; 5Department of Microbiology, Multidisciplinary Research Center on Antimicrobial Resistance, “Victor Babes” University of Medicine and Pharmacy, Eftimie Murgu Square No. 2, 300041 Timișoara, Romania; 6Department of Epidemiology, Infectious Diseases, Preventive Medicine and Epidemiology, “VASILE GOLDIŞ” Western University of Arad, Revolutiei Boulevard, No. 94–96, 310025 Arad, Romania; 7National Public Health Institute, “Victor Babes” Boulevard, No. 16, 300226 Timișoara, Romania; 8Department of Physiology and Pathophysiology, Medicine Faculty, “VASILE GOLDIŞ” Western University of Arad, Revolutiei Boulevard, No. 94–96, 310025 Arad, Romania; 9MedLife HyperClinic, Eroilor de la Tisa Boulevard, No. 28, 300551 Timișoara, Romania

**Keywords:** AIDS, HIV-1, subtype F1, CKD, survival

## Abstract

Human immunodeficiency virus (HIV) is a lentivirus that is transmissible through blood and other body fluids. During the late 1980s and early 1990s, an estimated 10,000 Romanian children were infected with HIV-1 subtype F nosocomially through contaminated needles and untested blood transfusions. Romania was a special case in the global acquired immunodeficiency syndrome (AIDS) pandemic, displaying the largest population of HIV-infected children by parental transmission between 1987–1990. In total, 205 HIV-infected patients from the western part of Romania were analyzed in this retrospective study. Over 70% of them had experienced horizontal transmission from an unknown source, while vertical transmission was identified in only five cases. Most patients had a moderate to severe clinical manifestation of HIV infection, 77.56% had undergone antiretroviral (ARV) treatment, most of them (71.21%) had experienced no adverse reactions and many of those with HIV (90.73%) had an undetectable viral load. Renal impairment was detected in one third of patients (34.63%). Patients born before 1990, male patients, patients diagnosed with HIV before the age of 10, and those undernourished or with renal impairment had a shorter average survival time than the group born after 1990, female patients, patients receiving ARV treatment, patients with a normal body mass index (BMI) and those without renal impairment. Periodical monitoring of the estimated glomerular filtration rate (eGFR) level, as well as the detection of protein excretion, should be taken into consideration worldwide when monitoring HIV-positive patients; this in order to detect even asymptomatic chronic kidney disease (CKD), and to manage these patients and prolong their lives.

## 1. Introduction

Human immunodeficiency virus (HIV) is a lentivirus, part of the Retroviridae family, that is transmissible through blood and other bodily fluids. In can be transmitted through sexual contact, parenterally, perinatally or through breastfeeding. Soon after HIV infection, the patient develops a high viral load, which is detectable by the nucleic acid amplification of viral ribonucleic acid (RNA) or the detection of core protein p24 [1]. Only four to six weeks later, the body starts producing antibodies, and seroconversion is present by 3 to 6 months [2]. Usually, the primary HIV infection resembles flu-like symptoms or infectious mononucleosis and/or aseptic meningitis. Later on, HIV infection can be documented by the presence of HIV antibodies, which are detected in the plasma. After producing HIV antibodies, the HIV viral load continues to diminish, and patients can be asymptomatic for several years. Over time, the CD4+ lymphocite count declines due to viral destruction, apoptosis and CD8 lymphocyte activation. Cell-mediated immunity becomes compromised in time and exposes the patient to opportunistic infections, HIV-associated nephropathy, dementia and cancers [3]. The latter concludes in the HIV infection stage as acquired immunodeficiency syndrome (AIDS) [4].

Acquired immunodeficiency syndrome (AIDS) disease was recognized in 1981 and its cause, HIV infection, was identified in 1983. There are two types of HIV recognized: HIV-1, which is primarily hosted by chimpanzees, and HIV-2 primary, primarily hosted by sooty mangabey monkeys [5]. HIV-1 is found worldwide in humans and it can be genetically divided into groups M, N, O and P [5]. Each group has its own subtypes associated with particular geographical areas: subtype B is mainly present in America, Australia and western Europe, while Subtype C is found in southern Africa and India [5]. HIV-2 is similar to HIV-1, but causes milder forms of infection and is naturally resistant to non-nucleoside reverse transcriptase inhibitors (NNRTIs) [6].

During the late 1980s and early 1990s, an estimated 10,000 Romanian children were infected with HIV-1 subtype F nosocomially through contaminated needles and untested blood transfusions [7,8]. Phylogeographic analysis of the pandemic lineage of HIV-1 group M suggests that the origins of this pandemic lie in Central West Africa. Group M is classified into nine subtypes (A–D, F–H, J and K). Out of the nine subtypes, subtype F causes less than 1% of all HIV-1 infections globally [5]. However, more than 99% of sampled HIV infections in Romania descend from a major lineage within subtype F, known as F1 [9]. Subtype F1 sequences have been sampled in several European countries, including Russia, Bulgaria and Belgium, even though Romania has the highest prevalence of subtype F in Europe (>70%) [5,8,10]. In Romania, subtype F1 seems to have been disseminated by the heterosexual route among the adult population and by the parental route in children, via the re-use of unsterilized needles and syringes and/or the transfusion of unscreened blood or blood products [10,11]. Romania was a special case in the global AIDS pandemic as it displayed the largest population of HIV-infected children by parenteral transmission between 1987–1990 [10,11].

The particularities of the Romanian HIV-infected cohort, the HIV-1 subtype F1 prevalence and the large pediatric population make this cohort rather unique among European cohorts. Patients from the Romanian cohort have been treated and exposed to antiretroviral (ARV) drugs in all available therapeutic combinations, even before the Highly Active Antiretroviral Therapy (HAART) era [12]. In Romania, ARV treatment was available starting from 1995, initially as monotherapy with one nucleoside reverse transcriptase inhibitor (NRTI); then, between 1996 and 1998, double therapy (two NRTI) was available, and starting from 1998, HAART was available, consisting of two NRTI and one protease inhibitor (PI). The detection of the viral load was possible only after 2001 [12]. Given the unique background and a history of over 20 years of HIV infection in patients, we performed a study so as to evaluate participants for renal impairment in order to raise awareness about the disease and its influence on the survival rates of children infected with HIV.

## 2. Materials and Methods

### 2.1. Study Design and Patients

We performed a retrospective study of 205 patients infected with HIV, from the Romanian HIV-infected cohort. The study was conducted in the western part of Romania, where patients had been admitted from five counties to the Emergency Clinical Hospital for Children in Timișoara.

### 2.2. Laboratory Results, Anthropometric and Clinical Assessment

We collected demographic data (age, gender, weight, height of patient at the time of HIV infection diagnosis), epidemiological parameters (mode of HIV transmission), and laboratory parameters (viral load (HIV RNA), CD4 cell counts, serum creatinine, serum urea, urine analysis), as a standard of clinical care. Data were obtained from the medical observational charts of patients admitted to the Infectious Diseases and Nephrology Department at the Emergency Clinical Hospital for Children “Louis Turcanu” Timisoara.

Informed consent was obtained in order to take 2 mL of venous blood, which was placed into tubes containing EDTA. Then, 1 mL of the whole blood sample was used to establish the full blood counts, lymphocyte differential counts, and specific CD4+ lymphocyte count measurements. The method used was standard flow cytometry. We measured CD4+ lymphocytes on the same day of blood collection using flow cytometry (FACSCAN analyzer, Becton Dickinson, Oxford, UK) with standard methods.

Whole blood was collected in EDTA tubes and centrifugated for 10 min at 3000 rpm. The resulting plasma was separated and used for viral RNA extraction. In order to obtain good-quality plasma samples for further processing, the whole blood tubes were not frozen.

HIV RNA extraction was performed according to the manufacturer’s instructions using a Maxwell^®^ RSC Viral Total Nucleic Acid Purification Kit.

The real-time amplification of isolated viral RNA was performed on the Biorad CFX96 Real-Time System with a method that permitted the detection of HIV RNA blood plasma (viral load) using specific primers and probes with two fluorophores, FAM and JOE. The amplification program had a reverse-transcription step, followed by 45 amplification cycles (denaturation, primer attachment and elongation).

We recorded all viral load and CD4 count data throughout the study period and analyzed the viral load (number of copies of HIV RNA/mL) and CD4 cell count (expressed as cells per cubic millimeter) in relation to weight, height, Body Mass Index (BMI), age at diagnosis, whether the patient had chronic kidney disease (CKD) or other renal manifestations, and the estimated Glomerular Filtration Rate (eGFR).

The serum creatinine value was measured at the central laboratory of the Emergency Clinical Hospital for Children “Louis Turcanu” using the Jaffe method until 2014 and the enzymatic method after 2014. Median serum creatinine values were reported in milligrams per deciliter. Creatinine clearance and eGFR were calculated using the Schwartz Formula.

BMI was calculated at the time of diagnosis with a weight/height (meters)^2^ formula. The results were adapted and interpreted for children according to their BMI for age charts in patients under 18 years.

We defined CKD regarding the eGFR in the following: stage 1 (≥90 mL/min/1.73 m^2^), stage 2 (60–89 mL/min/1.73 m^2^), stage 3 (30–59 mL/min/1.73 m^2^), stage 4 (15–29 mL/min/1.73 m^2^), and ESRD or stage 5 (≤15 mL/min/1.73 m^2^).

### 2.3. Statistical Analysis

Data were collected and analyzed using the SPSS v.17 software suite (SPSS Inc., Chicago, IL, USA) and are presented as mean ± standard deviations for continuous variables with Gaussian distribution or percentages for categorical variables. Survival was analyzed using the Hazard Ratio (HR) method and presented using Kaplan–Meier diagrams.

In order to assess the significance of the differences between groups, the student *t*-test (means, Gaussian populations), Chi-square (proportions) and log-rank test (differences between survival curves and hazard ratio) were used. Continuous variable distributions were tested for normality using D’Agostino’s and Pearson’s tests, and the equality of variances was tested using Levene’s test. For evaluating the involvement of more confounding factors in dichotomous outcomes, multivariate backward conditional (stepwise; the threshold for including a predictor in the model was established at *p* = 0.10) logistic regression models were built; their goodness of fit was estimated using the Hosmer–Lemeshov method. A *p* value of <0.05 was considered as the threshold for statistical significance.

### 2.4. Univariate Analysis

A Kaplan–Meier survival analysis was conducted to compare the survival distributions among HIV-positive patients considering their birth period, gender, age and BMI category for diagnostics, ARV treatment, as well determining the development of renal affection.

### 2.5. Patients Consent and Ethics Statement

The study was conducted in accordance with the Declaration of Helsinki, and approved by the Institutional Review Board (or Ethics Committee) of the Emergency Hospital for Children ‘Louis Turcanu’ Timisoara (93/2020).

Informed consent was obtained from all subjects involved in the study.

## 3. Results

In total, 205 HIV-infected patients were included in this retrospective study, all of whom were admitted to the Immunodeficiency Department of the Emergency Clinical Hospital for Children “Louis Turcanu” Timisoara between 1990–2022 from four counties in the western part of Romania.

### 3.1. Age, Sex, Urban/Rural Area Distribution

In the study cohort, 60.48% were females aged between several days and 22 years old at their first admission, of whom 57.07% were diagnosed at less than 10 years of age and 5 were infants with an age < 1 year. Initially, the study started with children born mainly between 1988 and 1991, belonging to the Romanian cohort of horizontally HIV-infected children, but we also included HIV-infected children born from HIV-positive mothers via vertical transmission.

Most patients were from the urban areas (171, 83.41%) of five counties in the western part of Romania.

### 3.2. Source of Infection

The source of HIV infection was unknown for 70.73% of patients, 10.73% of patients were infected via blood contamination, 8.29% were infected through the infusion of gamma globulins and 7.8% had experienced infection related to medical devices. Vertical transmission was observed in only 5 cases (2.43%) (Table 1), all of them after 2010.

### 3.3. Body Mass Index (BMI)

The participants’ BMIs ranged between 8 and 30 kg/m^2^ (mean 17 ± 4.21); we adapted the BMI values for children less than 13 years according to the national BMI for age percentiles. In total, 135 (65.85%) patients were considered undernourished, weighing less than the fifth percentile for age, height and BMI according to growth charts; 66 (32.19%) were at a normal weight; and 4 patients were overweight, with a weight that was above the 95th percentile for age, height and BMI in the growth charts.

### 3.4. HIV Infection Stage

Most patients (144; 70.29%) had moderate to severe clinical manifestations of HIV infection (class B or C). In total, 110 patients (53.65%), which is more than half, were on combined ARV treatment (2NRTI + 1PI), while 32 (15.60%) were on 2NRTI + 1NNRTI, and only 10 (4.87%) and 7 (3.41%) were on 2NRTI and 1NNRTI + 1PI, respectively. In total, 46 patients were either without treatment or were missing data regarding compliance to treatment.

Most of the patients (146; 71.21%) had no adverse reactions to medication, while 40 (19.51%) patients presented lipodystrophy dyslipidemia. Other minor adverse reactions were noted, such as fatigue, unusual prolonged tiredness or a loss of appetite.

HIV viral load was undetectable (<50 copies/mL) in 186 (90.73%) patients at the time of inclusion; in the remaining patients, it varied between 431 and 5,011,872 copies/mL (mean 74,283.72 ± 431,273.25). Only 71 (34.63%) had a normal CD4+ T-lymphocyte count, and most patients (134; 65.36%) had a CD4+ T-lymphocyte count < 200 cells/mm^3^.

### 3.5. Renal Impairment

More than a quarter of patients (71; 34.63%) presented a stage of CKD, i.e., 32 (15.60%) patients were in the first stage, 27 (13.17%) patients were in the second stage, while 8 (3.90%) and 2 (0.97%) patients were in the third and each of the fourth and fifth stages, respectively. The presence of CKD was significantly associated with diagnostic age (*p* = 0.045), but it was not significantly associated with weight, height, BMI, VL or CD4+ T-lymphocyte count. Table 2 presents the HIV-infected sample’s characteristics among the patients with and without CKD.

Diagnostic age was higher among patients with CKD (*p* = 0.045), while weight, height and BMI did not differ among patients with and without CKD (*p* > 0.05). Additionally, HIV VL and CD4+ T-lymphocyte counts did not differ among patients with and without CKD (*p* > 0.05).

Hematuria was identified in 21 patients, 14 of whom presented moderate to severe clinical manifestations of HIV infection (class B or C) (Table 3). 

The eGFR varied between 7 and 274 mL/min/1.73 m^2^, with a mean of 112.62 ± 42.77 mL/min/1.73 m^2^. Glomerular hyperfiltration (eGFR > 140 mL/min/1.73 m^2^) was observed in 50 (24.39%) patients. Absolute values of eGFR were not different among patients on ARV treatment or not (*p* = 0.271). The eGFR was not associated with the clinical stage of HIV infection (*p* = 0.213), age (*p* = 0.073), or the CD4+ T lymphocyte count (*p* = 0.653). Serum creatinine varied between 0.26 and 9.72 mg/dL, with a mean of 0.77 ± 0.79 mg/dL.

### 3.6. Survival Analysis

Patients born before 1990 had an average survival time of 17.16 (95% CI, 5.73 to 28.591) years, ranging between a few months and 33 years. This was shorter than the group of patients born after 1990, whose mean survival time of 18.692 (95% CI, 5.771 to 31.613) years was higher; this ranged between 1.5 years and 33 years. The log rank (Mantel–Cox) test was conducted to determine whether there were differences in the survival distributions for the different groups of patients. The survival distributions were not statistically significantly different, being χ^2^(1) = 0.261, *p* = 0.61. Female patients had a mean survival time of 18.769 (95% CI, 7.348 to 30.189) years, which was higher than for male patients, who had a mean survival time of 15.464 (95% CI, 3.092 to 27.836) years. The survival distributions of the two groups of patients were statistically significantly different, χ^2^(1) = 8.844, *p* = 0.003.

The malnourishment of HIV-positive patients was present in most of the events (92.98%), a normal BMI range in patients was present in only 7.02% of events, and no events were present in the group of overweight patients. The survival distributions of the groups of patients were statistically significantly different, χ^2^(2) = 28.977, *p* < 0.001. Patients with an HIV diagnostic age of less than 10 years had a mean survival time of 14.981 (95% CI, 3.007 to 26.955) years, which was lower than patients with a higher diagnostic age, who had a mean survival time of 21.487 (95% CI, 10.411 to 32.563) years. The survival distributions of the two groups of patients considering their diagnostic age were statistically significantly different, χ^2^(1) = 29.688, *p* < 0.001. Patients who followed an ARV treatment had a mean survival time of 19.918 (95% CI, 8.558 to 31.279) years, which was higher than for patients with no ARV treatment, who had a mean survival time of 5.837 (95% CI, 3.063 to 8.61) years. The survival distributions of the two groups of patients were statistically significantly different, χ^2^(1) = 70.927, *p* < 0.001.

When studying the influence of the development of renal affection among HIV-positive patients, we considered the eGFR level as well as the renal stage. Patients with eGFR values lower than 30 mL/min/1.73 m^2^ presented a significantly decreased mean survival time of 4 (95% CI, 0.88 to 7.12) years, which was much lower than patients with higher eGFR values, who had a mean survival time of 18.721 (95% CI, 7.304 to 30.138) years. The survival distributions of the two groups of patients were statistically significantly different, χ^2^(1) = 31.894, *p* < 0.001. Patients at the fourth and fifth stages of renal affection presented the lowest mean survival time, namely 3.5 (95% CI, 0.5 to 2.52) and 8 (95% CI, 2.12 to 13.88) years, respectively. The survival distributions of the groups of patients were statistically significantly different, χ^2^(5) = 17.191, *p* = 0.004. Pairwise log rank comparisons were conducted in order to determine which groups had different survival distributions. A Bonferroni correction was made, with statistical significance accepted at the *p* < 0.01 level. There was a statistically significant difference in the survival distributions for the HIV-positive patients with normal to mildly reduced kidney function vs. patients with severely reduced kidney function, χ^2^(1) = 8.457, *p* = 0.004, and χ^2^(1) = 9.912, *p* = 0.002, respectively, and hypnotherapy vs. e-cigarette intervention, χ^2^(1) = 29.003, *p* < 0.0005. However, the survival distributions for the severely and very severely reduced kidney function were not statistically significantly different, χ^2^(1) = 2.882, *p* = 0.09.

#### Multivariate Cox Regression Model

The hazard ratio (HR) for BMI was 0.747, so an increase in BMI of 1.0 kg/m^2^ was associated with a 0.747-fold increase in the hazard. The HR for a diagnostic age of less than 10 years was 3.373, so HIV-positive patients diagnosed at an age of less than 10 years presented a hazard that was 3.373 greater than that for patients diagnosed at an older age. In addition, HIV-positive patients with eGFR values smaller than 30 mL/min/1.73 m^2^ presented a HR that was 3.257 greater than that for patients with higher eGFR values. Moreover, HIV-positive patients not undergoing ARV treatment had a HR that was 4.394 greater than those following at least one class of ARV treatment. Statistical analysis showed a superior survival rate among those HIV-positive patients with normal and mildly reduced kidney function.

We mention that most of the HIV-positive patients diagnosed at a very low age were found undernourished. A negative coefficient for BMI (B= −0.292) indicated that lower values were associated with a higher risk and consequently a shorter survival time.

We found that HIV-positive patients diagnosed at an age less than 10 years presented a greater risk and, therefore, a shorter survival time (B coefficient = 1.216).

In 2022, only 32 patients were still in monitoring due to a lack of follow-up, administrative restructuration, a lack of treatment adherence or an undesired outcome. Out of 32 patients, 37.50% were males and 62.50% came from urban areas. The median age was 32. Most of patients (81.25%) had a CD4 level above 200/mm^3^ and 68.75% had undetectable HIV VL. Only one patient was diagnosed with CKD stage II (eGFR of 62 mL/min), while 50% of patients had modified urinalysis (hematuria, leukocyturia or some grade of proteinuria).

## 4. Discussion

Our study described the incidence of renal impairment and its manifestations in a Romanian cohort of patients infected with HIV. The Romanian cohort was mostly infected by subtype F1 HIV [7,10], with over 74% predominance [13]. A high prevalence of subtype F1 is also present in other countries, ranging from 8% to 23% in HIV-positive patients in Angola, and from 10 to 20% in HIV-positive patients in Brazil [10]; it was also identified in a small cluster in Spain. Throughout Europe, subtype F is relatively rare, and is mostly encountered in the Romanian HIV cohort of patients, where this subtype reaches a prevalence of over 70% [8,11]. In Europe, South and North America, Australia, the Middle East and North Africa, subtype B is still the most widespread viral strain. Subtype C has been reported among HIV/Viral hepatitis patients in Southern Brazil and Asia, while Subtype A predominates in India, Russia and other former Soviet Union countries [8,14]. Several recombinant strains have been identified: AD and AC recombinants have been described in Eastern Africa and India, and the BF recombinant strain has been reported in Brazil and Argentina [14]. Therefore, the HIV-positive Romanian cohort consisting mainly of children with HIV subtype F1 via horizontal infection is particular [13]. Recognizing the main subtype of HIV involved in each region is important when comparing data in the present study with findings in the literature.

### 4.1. Age, Sex, Urban/Rural Distribution

In the study cohort, more than half of the patients (60.48%) were females and 57.07% were diagnosed with HIV infection before the age of ten. Most patients were from urban areas (83.41%). Other studies on HIV-infected children have shown a similar gender distribution (58.51% females) and urban area predominance (90.52%) [15].

### 4.2. BMI

In our study, 135 (65.85%) patients were considered undernourished at the time of diagnosis.

In HIV-infected children, undernutrition is a major public health problem that limits the desirable success of HAART, particularly in developing nations [15]. Undernourishment was noted in between 44.51% and 51.62% of pediatric patients in previous studies in low-income countries [15,16]. A high magnitude of undernourishment at the baseline of ARV treatment might be due to a high viral replication that might cause an immune-dysfunction that leads to susceptibility to infectious diseases and the malabsorption of micronutrients [15].

### 4.3. Source of Infection

In our study, the source of infection was unknown for 70.73% of patients, while only 10.73% were a result of untested blood transfusions, 8.29% were a result of gamma globulin transfusion and 7.8% were a result of medical device-related infection. Only five patients in the study had documented vertical transmission. The unknown cause of infection might due to the lack of accurate data in the 1988–1990 period due to the socio-political conditions in the country [17]. The horizontal transmission of HIV through the improper use of needles and syringes is strongly suspected [11,17].

Children generally acquire HIV infection through vertical transmission occurring in utero, during labor and delivery, or through breastfeeding after birth. The ARV treatment of second trimester HIV positive mothers reduces the risk of vertical transmission from 15–25% to below 5% [13]. Earlier studies have reported the administration of contaminated blood products, sexual abuse, surrogate breastfeeding, the reuse of unsterile needles, and premastication as the modes of nonvertical HIV transmission in children [18]. However, the identification of the source of infection and the route of transmission is not possible in every case [3,10,11,17].

### 4.4. Renal Impairment

The incidence and prevalence of CKD simultaneous with HIV infection varies across geographic areas, with large differences also within the same continent. This variability depends on multiple factors: the methods used to evaluate renal function, the definition of CKD, genetic heterogeneity, prevention programs, access to the health care system and the initiation of combined ARV treatment [19].

In this study, we identified 71 patients (34.63%) that presented a stage of CKD, which was relatively higher compared to other studies; this is probably because of the characteristics of the patients, horizontal transmission and possibly a subtype of HIV. The estimated prevalence of CKD in the HIV-positive population—measured using a CKD–EPI equation—ranges from 2.51% in Europe to 7.44% in North America [19]. A Danish HIV cohort reported a CKD prevalence of 6% in HIV-infected patients [20], a prevalence of 24% in Nigeria in antiretroviral-naïve HIV-positive patients [21], and a CKD prevalence that ranged between 2% [22], 10,22% [23] and 15.51% [24] in an American HIV-infected population. This great range depends on the study design, the HIV-positive population included in the study and the ARV treatment. In particular, Stage 2 CKD was found in 13.17% of patients included in this study, which is lower compared to other studies: 29.41% in a Bucharest study [12], 16.58% in a West African HIV-infected population [25] and 15.31% in a Nigerian antiretroviral-naïve HIV-positive population [21]. Stage 3 CKD was found in 3.90% of patients in our study, which is a higher percentage compared to the study in Bucharest, Romania, in which 1.42% of HIV-infected patients had stage 3 CKD [12]. In a very similarly designed study, regarding Brazilian HIV-infected children and adolescents, none of the patients were diagnosed with CKD, as the eGFR calculated by the Schwartz formula varied between 89 and 256 mL/min/1.73 m^2^ [26].

The glomerular hyperfiltration status, defined as eGFR over 140 mL/min/1.73 m^2^, was identified in our study in 24.43% of patients. In total, 82% of the patients included in the Brazilian study experienced glomerular hyperfiltration [26]. Glomerular hyperfiltration is encountered in HIV infection, diabetes and arterial hypertension. HIV, in particular, infects epithelial cells and disturbs podocyte structure and function, leading to an up-regulation of the renin-angiotensin system, which is the basic pathophysiology of hyperfiltration [27]. In addition, HIV by itself, ARV treatment and metabolic syndrome may lead to hyperfiltration due to insulin-like growth factor [28]. Similar data, of a 24% hyperfiltration status, was found in a cohort of HIV-infected children in Benin, which was more frequently encountered in children receiving ARV treatment [29].

In the present study, serum creatinine varied between 0.26 and 9.72 mg/dL, with a mean of 0.77 ± 0.79 mg/dL. Compared to the data in the literature, several studies on HIV-infected children report mean creatinine levels of 0.81 ± 0.3 mg/dL in HIV–CKD pediatric patients, 0.6 ± 0.18 mg/dL in HIV-positive children with intermittent proteinuria and 0.57 ± 0.2 mg/dL in HIV-positive children without proteinuria [30]. Our data seem closer to the HIV–CKD pediatric patients’ values. The American study aforementioned remarks that most of its patients were of African descent (89%) and had acquired HIV-1 via vertical transmission [30]; meanwhile, subtype B, group M HIV is predominant in the American area [6]. It remains unclear whether infection with subtype F1 of HIV represents a risk factor for developing CKD by itself.

### 4.5. Survival Analysis

We observed that the survival distribution of HIV-infected patients was statistically significantly influenced by gender (*p* = 0.001), BMI (*p* = 0.002), the diagnosis of HIV infection at an age less than 10 years (*p* < 0.001), the presence of ARV treatment (*p* < 0.001) and eGFR values lower than 30 mL/min/1.73 m^2^ (*p* = 0.004), which are associated with moderate to severe stages of CKD.

#### 4.5.1. Survival Influenced by Gender

In the literature, the female sex was associated with an increased risk of death when analyzing the HIV RNA level [31], but when correlating state income with gender and life expectancy, the data suggest that there are no gender differences in life expectancy in high-income countries; however, life expectancy is consistently higher in women than men in low/middle-income countries [32]. In the present study, female patients had a longer mean survival time and a lower HR compared to male patients.

#### 4.5.2. Survival Influenced by BMI

We mention that most of the HIV-positive patients diagnosed at a very low age were found undernourished. A negative coefficient for BMI (B= −0.292) is associated with a higher risk of events and consequently a shorter survival time.

Other studies on children infected with HIV have revealed that the mean survival time for children who were undernourished at the time of ARV treatment initiation was 50.6 months (SD ± 1.7), and 61.1 months (SD ± 0.6) for those who were well nourished [16]. Undernutrition, together with HIV infection, negatively affects immunity, leading to severe immune dysfunction and susceptibility to infectious diseases [15].

#### 4.5.3. Survival Influenced by Age of Diagnosis

We found that HIV-positive patients diagnosed at an age less than 10 years presented a greater risk of complications and therefore a shorter survival (B coefficient = 1.216).

#### 4.5.4. Survival Influenced by ARV Treatment

In our study, as expected, patients with no ARV treatment presented a greater risk of events and consequently a shorter survival. Life expectancy after starting combined ARV treatment improved over time in both high- and low/middle-income countries [32]; as such in 2016, the World Health Organization (WHO) published a correction report that moved the ranking of HIV infection/AIDS from the second leading cause of adolescent death down to eighth place. However, these changes were largely not due to actual improvements in HIV infection diagnosis or starting children and adolescents on life-saving ARV treatment sooner, but because of lower rates of mother-to-child HIV transmission for different categories of women at risk (e.g., from 30% to 18% if seroconverting during pregnancy) and efficient ARV prophylactic treatment in the newborn [33].

#### 4.5.5. Survival Influenced by CKD

CKD is often asymptomatic and is found during routine screening. Guidelines recommend that patients with HIV infection should be screened regularly for CKD to allow for its early detection and adequate management [34]. In our study, values of eGFR lower than 30 mL/min/1.73 m^2^ were associated with a greater risk of CKD and therefore a shorter survival time (B coefficient = 1.181).

The life expectancy of HIV-infected patients with impaired renal function has improved significantly after the introduction of HAART [35]. Although HAART is an important part of treatment, the progression to end-stage kidney disease still remains a significant co-morbidity [35].

International recommendations suggest that patients with HIV infection should have their GFR estimated at least every 6 months and should have either a urinalysis or a quantitative assessment of their urinary protein excretion performed at least annually [30,34,36]. In particular, those whose eGFR has declined by 25% or more, or to a level below 60 mL/min per 1.73 m^2^, or who have a protein excretion of over 300 mg/day, should be referred for evaluation by a Nephrologist for further management [34].

## 5. Conclusions

The periodical monitoring of the eGFR level, as well as the detection of protein excretion, should be taken into consideration worldwide when monitoring patients with HIV infection, so as to detect asymptomatic CKD earlier, and manage CKD and assure better survival outcomes.

## Figures and Tables

**Table 1 life-13-00888-t001:** Patient clinical classification and modality of HIV transmission.

Clinical Classification at Diagnosis ^1^	HIV Transmission					
	Unknown	Gamma Globulins	Medical Devices-Related Infection	Blood Products	Blood	Vertical
Class A	39 (19.02%)	9 (4.39%)	3 (1.46%)	2 (0.97%)	8 (3.90%)	0 (0%)
Class B	74 (36.09%)	5 (2.43%)	9 (4.39%)	1 (0.48%)	6 (2.92%)	3 (1.46%)
Class C	32 (15.60%)	3 (1.46%)	4 (1.95%)	1 (0.48%)	4 (1.95%)	2 (0.97%)
Total	145 (70.73%)	17 (8.29%)	16 (7.8%)	4 (1.95%)	18 (8.78%)	5 (2.43%)

^1^ Classifying HIV infection into A, B, and C classes was used for the adult population and for children until 2007; however, starting from 2007, the four clinical classification stages of HIV infection for children aged over 15 were adapted and used, according to the WHO in 2007.

**Table 2 life-13-00888-t002:** Characteristics of the HIV-infected patients with and without CKD.

Variable ^1^	with CKD	without CKD
Age (years)	9.43 (±4.27)	8.17 (±4.44)
Weight (kg)	37.24 (±14.62)	36.72 (±17)
Height (cm)	143.13 (±23.44)	136.91 (±34.71)
BMI (kg/m^2^)	17.15 (±3.3)	16.91 (±4.63)
VL (copies/mL)	41,960.55 (±172,379.86)	91,410.17 (±518,454.82)
CD4+ T-lymphocyte count (cells/mm^3^)	234.59 (±293.3)	201.4 (±283.16)

^1^ Values of variables are presented as mean (±standard deviation).

**Table 3 life-13-00888-t003:** Frequency of renal manifestations in HIV-infected patients.

Renal Manifestations *	Class A	Class B	Class C
Urinary tract infection (UTI) or recurrent UTI	43 (21.6%)	80 (40.2%)	38 (19.1%)
Renal lithiasis	8 (4%)	15 (7.5%)	4 (2%)
Nephritic syndrome	1 (0.5%)	3 (1.5%)	0 (0%)
Acute renal failure	1 (0.5%)	5 (2.5%)	5 (2.5%)
Chronic renal failure	3 (1.5%)	3 (1.5%)	1 (0.5%)
Hematuria	7 (3.5%)	13 (6.5%)	1 (1.5%)
Enuresis	2 (1%)	3 (1.5%)	3 (1.5%)
Malformations	2 (1%)	2 (1%)	5 (2.5%)

* Most patients presented combined renal manifestations.

## Data Availability

Not applicable.

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
