# Peer review of "Renal Impairment Impact and Survival Analysis in a Romanian Cohort of HIV-1(F1)-Infected Children and Adolescents"

_life, 2023, doi:10.3390/life13040888_

Round 1

Reviewer 1 Report

The authors summarized the findings on the role of monitoring HIV positives patients in an early detection of asymptomatic CKD, which makes it better manage to survive. However, there are many errors or fixes in the manuscript. Below are the  examples: 

1. The abbreviations AIDS, BMI, eGFR, and CKD are not defined in the abstract. 

2. Lines 44-45. This sentence should be revised.

3. Line 46. RNA not ARN. 

4. Line 52. CD4+ lymphocyte.

5. Line 65. Reverse transcriptase.

6. Line 137 and others. HIV infected/positive patients not HIV patients.

7. Lines 172 &173. PI not IP.

8. Line 171 & others. ART and ARV treatment are mixed in the manuscript.

9. Line 200 & others. HIV infection not HIV.

10. Line 206 & others. mg/dL and mg/dl are mixed.

11. Line 209. Several not few.

12. Line 267. mm3 not mmc.

13. Line 290. From not form.

14. Line 358. Area not aria.

15. Line 371. BMI not MBI.

16. References.  Please check instructions for the authors.

Author Response

Dear reviewer, thank you for your kind observations. We improved the manuscript as suggested. Please see the attachment.

Best regards!

Reviewer 2 Report

In this manuscript, Isac et al. performed a retrospective analysis of 205 Romanian HIV-infected patients with a focus on chronic kidney disease (CKD). From their statistical analyses, the authors found that ~35% of patients presented with chronic kidney disease (CKD), which was significantly associated with the age of diagnosis. Moreover, they found that severe CKD was associated with reduced survival time. This is an interesting study that complements other studies regarding renal impairment in HIV-infected patients. It also provides information for other patient parameters. While this study is interesting and important to the HIV field, the presentation of continuous variables as fixed means and categorical variables as fixed percentages provides limited information. For example, during the course of patient evaluation, it is impossible to know patients’ progression from class A to B or from B to C or changes in patients’ CD4 T-cell counts. Another major concern is that the authors did not analyze CKD with respect to ART. In addition to CKD caused by HIV infection of renal epithelial cells, ART might affect the kidney function (earlier ART may have produced more renal toxicity than recent ART). Some additional comments are as follows:

The patients in this study were infected with subtype F, which is not common to Western or Eastern Europe; however, it is unclear whether subtype F has any bearing on the results.

Usually for studies involving patient specimens, Materials and Methods have a description/statement regarding patient consent as well as an ethics statement.

Most statements are only referenced with one citation even though there may be multiple published studies to cite.

Please describe/reference methods used to measure CD4 counts and HIV RNA.

The background history of the nosocomial infections in Romania is confusing in the context that the source of HIV infection was unknown for 70.73% of patients in the study, most apparently from the 198-1991 era.

It is difficult to see the significant association of CKD with age but not other attributes in Table 2 (may relate to the major concern above).

Author Response

Dear reviewer, thank you for your remarks. We improved the manuscript as suggested.

Best regards!
